# High-Throughput Preparation and Characterization of ZrMoTaW Refractory Multi-Principal Element Alloy Film

**DOI:** 10.3390/ma15238546

**Published:** 2022-11-30

**Authors:** Qiannan Wang, Hongwang Yang, Xiaojiao Zuo, Yinxiao Wang, Jiahao Yao

**Affiliations:** 1School of Materials Science and Engineering, Shenyang University of Technology, Shenyang 110870, China; 2Beijing Institute of Technology Chongqing Innovation Center, Chongqing 401120, China

**Keywords:** refractory multi-principal element alloys, high-throughput screening technology, multi-gradient deposition method

## Abstract

In this work, high-throughput screening technology is applied to four-member refractory multi-principal element alloys (RMPEAs) films with high W content. The exploration of refractory metals such as W is strictly limited by the high melting temperature in this work; a multi-gradient deposition method was introduced to overcome this obstacle. By adjusting the power and distance from the target to the sample, component Zr_11_Mo_11_Ta_25_W_53_ with the best hardening performance was successfully obtained. The uniformity of the material library was analyzed from the perspectives of phase structure and micromorphology. With the help of Hume-Rothery theory and XRD analysis, it is shown that the film has a stable bcc structure. It is believed that film uniformity, nanoscale size, preferential orientation, surface roughness, and solution mechanism are the pivotal factors to improve hardness performance, especially for high W components. The hardness and modulus of elasticity can reach 20 GPa and 300 GPa, respectively, and the H/Er and H^3^/Er^2^ values are 0.067 and 0.065, showing the best wear resistance in many samples.

## 1. Introduction

High-entropy alloys (HEAs) were originally proposed by Yeh in terms of chemical composition [1], containing at least five or more elements and mixing them with the same atomic percentages. At present, the study of HEAs is not limited by the original definition, but has expanded it to “complex concentrated alloys (CCAs)” or “multi-principal element alloys (MPEAs)” [2,3,4].

A high-entropy alloy with more than three refractory metals is called a refractory high-entropy alloy (RHEAs) [2]. RHEAs are considered as the HEAs based on refractory elements, such as Mo, V, Hf, W, Cr, Ta, Nb, Zr, and Ti, and the refractory elements are mostly BCC structures. RHEAs exhibit excellent properties, such as strong corrosion resistance, high strength, good ductility, and good wear resistance, especially high hardness [5,6,7,8,9,10,11,12,13,14,15]. It is reported that the maximum strength of refractory high entropy alloy breaks through the strength limit of traditional refractory alloys. Senkov et al. [10,16] reported both VNbMoTaW and NbMoTaW alloys with a single-phase BCC structure, and the room temperature compressive yield strength (σ_y_) was 1058 MPa and 1246 MPa, respectively. The equal atomic ratio TiZrHfNbTa alloy also has a body-centered cubic (BCC) structure [16], the compressive yield strength σ_y_ is 929 MPa, and the compressive plastic strain can exceed 50%. Wu et al. [17] reported a novel Hf_25_Nb_25_Ti_25_Zr_25_ RHEA. The RHEA has a single BCC structure with breaking strength and plastic strain reaches about 969 MPa and 14.9%, respectively. TiZr_0.5_NbCr_0.5_V, TiZr_0.5_NbCr_0.5_, and TiZr_0.5_NbCr_0.5_Mo alloys with excellent corrosion resistance also have a BCC structure [18].

The traditional “trial and error” method makes minor adjustments to finite element types, which seriously affects the efficiency of research. Regarding the improvement of efficiency, researchers introduced magnetron sputtering technology in MPEAs and tried to explore a variety of rapid screening methods. Magnetron sputtering, as a typical physical vapor deposition (PVD) technique, can prepare high-quality, completely dense, and well-adhered films. It has been widely used to prepare multi-principal alloys such as AlCrTaTiZr [19], AlCoCrCu_0.5_FeNi [20], BiFeCoNiMn [21], and (TiAlCrNbY)C [22]. Chen et al. [19] first prepared FeCoNiCrCuAl_0.5_ and FeCoNiCrCuAlMn nitride films by magnetron sputtering. Zhang et al. [23] used multi-target co-sputtering to obtain gradient materials for Ti-Al- (Fe, Ni, Cr) alloys. Liu et al. [24] used magnetron sputtering to obtain 5700 XRD diffraction patterns from 12 alloy thin film systems. By analyzing an unprecedented amount of experimental data, it was found that a larger full-width at half-maximum (Δ q) of the higher glass formation ability (GFA) was related to the first diffraction peak in the XRD diffraction pattern. The alloy film obtained by thin film deposition technology has the advantages of controllable composition and structure, which broadens the way to synthesize RHEAs. Feng et al. [25] deposited TaNbTiW system films with different compositions and explained the structure and mechanical properties of the TaNbTiW films. The maximum hardness and modulus of elasticity obtained from multiple components are 5.2 GPa and 127.2 GPa, respectively. This high-throughput method can effectively reduce the number of bulk alloys prepared, reducing the high consumption of RMPEA in the metallurgical process. As a new refractory alloy system that has not been deeply explored, ZrMoTaW has the potential value of developing high-quality alloys. Moreover, the high melting point metal W content increases the difficulty of exploration. This work will attempt to explore the new alloy and provide a feasible method to facilitate research.

In this work, a novel body-centered cubic Zr-Mo-Ta-W RMPA film was proposed for the first time. The Zr-Mo-Ta-W thin film with gradient component library was prepared by multi-target magnetron co-sputtering. It has high hardness and high wear resistance. The structural stability, surface uniformity, and morphological characteristics of the RMPA were studied. The mechanism of enhancing the mechanical properties of this RMPA is discussed. At the same time, research on the later stage of the film also prospects.

## 2. Experimental Procedure

### 2.1. Multi-Gradient Material Library Preparation

The Zr-Mo-Ta-W quaternary alloy thin films in this work were deposited by a multi-target co-sputtering system (Shenyang Kecheng Vacuum Tech. Co., Ltd., Shenyang, China). Materials can be prepared by sputtering from four elemental targets. The targets are manufactured from hot-pressed elemental powders. The purity of four element targets (Shanghai Institute of Optics and Fine Mechanics, the Chinese Academy of Sciences) exceeded 99.9 percent. The dimensions are 60 mm in diameter and 3 mm in thickness. The center of the target is 100 mm away from the sample surface. The connection between the center of the target material and the center of the sample stage with the plane where the sample stage is located is at an angle of 45°. The film was grown using a commercial single crystal P-Si (100) with a diameter of 50 mm as a substrate, and the schematic diagram of the four-target co-sputtering of the experimental equipment is shown in Figure 1a. The base pressure of the deposition chamber was better than 9 × 10^−4^ Pa. The target is exposed to air and is prone to oxidation or contamination. To remove the effects of contaminants formed on the target surface, pre-sputtering must be performed at a working pressure of 0.4 Pa for 10 min prior to each deposition process. During the deposition of the film, the sample holder was water-cooled to maintain it at room temperature. The thickness of the film is kept above 1.5 μm by controlling the sputtering time and measuring it by profiler (Alpha-step IQ, KLA, Milpitas, CA, USA).

Performance-related high-throughput screening requires the preparation of multi-gradient material libraries by magnetron co-sputtering. A multi-gradient library of materials can be obtained by adjusting the power applied to the targets or by the distance of the target to different locations on the substrate. The tilt of the sputtering guns shown in the figure will cause the distance of the target to different locations on the substrate to be different, so that the incident flux is reduced from near the target to the position away from the target, thus forming a multi-gradient material library. The material library shown in Figure 1a is obtained by separating the alloys by a physical mask covered on the Si substrate. The dimensional design of the mask plate is shown in Figure 1b. There are 101 small round holes on the physical mask, each of which is treated as an independent sampling point of the material library, the so-called “sampling unit”. In other words, 101 independent sample units with completely different compositions can be obtained in one experiment. Table 1 provides the relevant experimental parameters during the preparation of the material library. The screening method provided in this experiment is significantly different from the traditional trial and error method, which may only obtain one alloy per day.

### 2.2. Film Characterization

The composition of the material library (Zr-Mo-Ta-W) was carried out using an energy dispersive X-ray spectrometer (EDS) equipped with field-emission scanning electron microscopy (SEM) (SUPRA 55 SEM, LEO, Oberkochen, Germany). The crystal structure of sample units was analyzed by using a glancing-incidence (1°) X-ray diffractometer (X-ray diffraction, XRD, Rigaku D/max-2400, Tokyo, Japan) using the Co Kα radiation at a scanning rate of 4°/min. The scanning step was 0.02°, and the scanning range was 30–90°. The hardness and elastic modulus of different sample units were determined by a nano-indentation test (NI, Agilent G200, Agilent Technologies, Santa Clara, CA, USA) which has a Berkovich triangular pyramid indenter. The tip was a circular arc with a radius of 20 nm. The specimen was pressed into with a continuously varying load to obtain a continuous load-displacement curve. To avoid the effect of the substrate on the hardness test of the material, the maximum displacement entering the sample surface was 300 nm, which was <10% of the film’s thickness.

## 3. Results and Discussion

### 3.1. Composition Gradient Analysis

The composition of obtained film was shown in Figure 2, as Zr, varies from 5 at.% to 55 at.%, Mo varies from 10 at.% to 50 at.%, Ta varies from 10 at.% to 50 at.%, and W varies from 10 at.% to 55 at.%. Figure 3 shows the contour plots of the change of Zr, Mo, Ta, and W content in the film. It was found that as long as the height and the sputtering angle between the target and substrate remained the same, a library of materials shows an unchanged trend of the composition gradient. First, the composition of HEA was predicted to be close to the intermediate region, and the first sample library region was selected as the film center region F6C. Then, we explored around a central area. In cases where the content of one element is significantly higher than that of the others, a change in the properties of the film can be observed. Even better sample libraries were explored. The specific components of the 6 sample libraries selected are listed in Table 1.

The corresponding thin film element distribution is shown in Figure 4. Taking sample F6C and sample D11 as an example, the composition of the whole film showed gradient changes, but the local element distribution was uniform. It confirmed local uniformity in the distribution of alloying elements. Khan et al. [26] demonstrated EDS mapping of films grown at different operating pressures and observed uniformity of film deposition. Sha [27] and Malinovskis et al. [28] obtained uniform film by rotating the substrate at a constant speed. In this work, using the four-target co-sputtering technique, it is still possible to obtain the alloy films with the locally uniformly distributed elements. Compared with many researchers such as Sha [27] and Malinovskis et al. [28], the multi-target co-sputtering technology can adjust the chemical composition more easily. Therefore, this technique is a very efficient method for preparing refractory multi-principal alloys (RMPAs).

### 3.2. XRD Analysis

The XRD pattern of the six samples is shown in Figure 5. According to the XRD patterns, all the samples exhibited BCC single-phase solid solution structure. This can be attributed to the following. 1. All the Zr, Mo, Ta, and W elements have the same BCC structure, and Mo-W, Ta-W, and Mo-Ta are completely mutually soluble [29,30,31]. W also has a certain solubility in Zr [32]. 2. The high entropy effect caused by multiphase mixing affects the stability of single-phase solid-phase solutions. The increase of entropy can effectively reduce the Gibbs free energy of the whole system and improve stability. This facilitates the intersolubility of elements to form simple single-phase structures instead of multiple phases. In other words, the decrease of Gibbs free energy can obviously reduce the tendency of order and segregation, making the solid solution easier to form and more stable than other ordered phases [33,34]. For the composition design of multi-principal alloys, predicting which elements will be more conducive to the formation of single-phase solid solutions and which elements are more thermodynamically stable when introduced are the first issues to be considered. According to the Hume-Rothery theory, the above issues can be effectively predicted. The empirical criteria are represented by the following:

The atomic radii and lattice constants of the elements are listed in Table 2. The lattice constant is estimated by a = 4r/√3. For multi-principal alloys, especially high-entropy alloys, all the principal elements have the same probability to occupy lattice sites to form a solid solution. Therefore, each component element in the alloy system can be regarded as a solute atom, and the serious lattice distortion caused by the large atomic radius difference between so many alloy elements makes the structure of the solid solution in RMPAs distinctive from that of pure metal and traditional alloys.

As can be seen from the Hume-Rothery rule, the difference between atomic radius δ can be expressed as [37,38]:(1)$$\delta =\sqrt{\overset{n}{\underset{i=1}{\sum }}c_{i}\left(1-\frac{r_{i}}{r\;}\right)²}$$
where *r_i_* is the atomic radius of the *i*th element, n is the number of elements,$\;c_{i}$ is the atomic percentage of one of the elements in a region of the films, $\bar{r}$ is the average atomic radius of the film. The average atomic radius ($\bar{r}$) [39] of the film:(2)$$\bar{r}=\overset{n}{\underset{i=1}{\sum }}c_{i}r_{i}$$

Thermodynamic parameter Ω [40] is further proposed:(3)$$\Omega =\frac{T_{m}△S_{mix}}{\left|△H_{mix}\right|}$$
(4)$$T_{m}=\overset{n}{\underset{i=1}{\sum }}c_{i}{\left(T_{m}\right)}_{i}$$
where *T_m_* is the theoretical melting point of an alloy, ${\left(T_{m}\right)}_{i}$ is the melting point of the *i*th component of alloy,

It has been reported that the conditions for the formation of a single solid solution phase are Ω≥1.1 and δ ≤ 6.6% [40]. Zhang et al. [37,38] also proposed that the mixing enthalpy ΔH_mix_ of the alloy in the liquid state can also be used as a criterion for solid solution formation:(5)$$△H_{\mathrm{mix}}=\sum_{i=1,i\neq j}^{N}4\Delta H_{ij}^{mix}X_{i}Y_{j}$$
In the equation, $\Delta H_{ij}^{mix}$ is the mixing enthalpy of the ith and jth components. When −15 ≤ ΔH_mix_
≤ +5 KJ/mol and δ ≤ 5%, it is beneficial to form a disordered solid solution.

Valence electron concentration (VEC) to evaluate the stability of the solid solution phase was proposed by Guo et al. [35].
(6)$$\mathrm{VEC}=\overset{n}{\underset{i=1}{\sum }}c_{i}{\left(VEC\right)}_{i}$$
where *VEC_i_* is the valence electron concentration of the ith component. When VEC ≥ 8.0, a single-phase FCC structural phase is formed; when the VEC ≤ 6.87, a single-phase BCC structural phase is formed; when 6.87 ≤ VEC ≤ 8.0, a duplex solid-solution phase of FCC and BCC structures is formed.

The parameters such as thermodynamic parameters (Ω), atomic size mismatch (δ), and valence electron concentration (VEC) of Zr-Mo-Ta-W RMPAs are listed in Table 3. For the Zr-Mo-Ta-W alloy system, thermodynamic parameters Ω≥1.1, δ ≤ 6.6%, and VEC ≤ 6.87, which satisfied the BCC solid-solution phase formation conditions. The mixed entropy (ΔS_mix_) obtained in the table is also significantly higher than that of conventional alloys. This means that the Zr-Mo-Ta-W RMPA films prepared in this experiment also have high mixed entropy, which helps to form a simple disordered solid-solution phase in the alloy system.

As the XRD diffraction pattern is shown, the (110) peak of the F10 film is higher, which indicates that the film exhibits an optimal orientation of (110). Some theories suggest that the grain surface grows slowly when the merit orientation is preferred. The relative degree of optimal orientation of the deposited coating can be expressed by the texture coefficient (TC), which is calculated as follows [41]:(7)$$\mathrm{TC}=\frac{I_{\left(hkl\right)}/I_{0\left(hkl\right)}}{\left(1/n\right)/\sum I_{\left(hkl\right)}/I_{0\left(hkl\right)}}$$
where $I_{\left(hkl\right)}$ is the peak strength of the film in the XRD diffraction pattern; $I_{0\left(hkl\right)}$ is the standard strength in the JCPDF database; n is the total number of diffractive surfaces. When the TC of the diffractive plane is unity, the crystal orientation distribution is random. When the TC of the (*hkl*) surface is greater than unity, it indicates that there is a merit-based orientation. The larger the value of TC, the greater the degree of merit orientation. The calculated TC values are listed in Table 4. From the TC value, E8 presents a (211) merit orientation, and most of the other units of the film present a (110) preference orientation, and in E8, the (110) degree of preference orientation is the greatest.

As shown in Figure 5b, (110) diffraction peaks shifted to a lower angle with increasing Zr from sample E8 to sample E4. When the XRD diffraction angle decreases, the surface spacing increases, which indicates that the lattice constant of the ZrMoTaW alloy increases [42]. The average atomic radius of each component is calculated using Equation (4) and the lattice constant of the different thin films is estimated by the equation a = 4ṝ/√3. Then, according to the relation of the Bragg Law 2dsinθ = nλ (λ = 0.179021 nm) and the interplanar spacing d_hkl_ = a/$\sqrt{h^{2}+k^{2}+l^{2}}$, the diffraction angles θ can be calculated. The calculated average atomic radius, lattice constant, interplanar spacing d_hkl_, diffraction angle θ (110), and d values of the samples E8, D11, I5, F6C, C5, and E4 are listed in Table 5. In the table, the average atomic radius is relatively small at 0.146 ± 0.003 nm, and the crystal plane spacing is maintained at 0.237 ± 0.005 nm. The Zr content increases and the corresponding mean atomic radius increases from 0.1432 nm to 0.1481 nm. Figure 6 shows the lattice constant and crystal plane spacing of the selected compositions. The content of Zr elements showed an upward trend, and its lattice constant and crystal plane spacing also showed a corresponding trend. Therefore, it is considered that in Figure 5b, the main reason for the small angular shift of the diffraction peak in the alloy ZrMoTaW is that the alloying elements with large atomic radii enter the lattice structure, causing lattice expansion and increasing in d value.

### 3.3. Surface and Cross-Sectional Morphology

The surface and cross-sectional morphologies of the ZrMoTaW quaternary thin film are shown in Figure 7. The Zr-Mo-Ta-W RMPA film with different compositions all exhibited a flake-like surface morphology. All the cross-section morphologies are typical dense columnar structures, as shown in Figure 7a–d. The grain sizes of the different components obtained in Table 6 according to the Scherrer equation:(8)$$D=\frac{k\lambda }{\beta cos\theta }$$
where *k* is the shape factor (*k* = 0.943 [43]), β is the full width at half-maximum. The grain size of the ZrMoTaW film is between 10 nm and 15 nm. Yeh et al. [1] suggested that diffusion for phase separation in HEAs is sluggish. The difficulty of displacement diffusion is affected by the elements in these alloys and the interaction between the formation and growth rates of atomic nuclei, which lead to the formation of ultrafine grains. On the other hand, high deposition rates also refine grains [44].

In Figure 2, sample I5 and sample E4 are close to the Mo target and Zr target, and sample E8 and sample C5 are close to the W target and Ta target. The deposition rate can be expressed in terms of deposition thickness per unit of time. Combined with the experimental parameters in Table 1 and the calculation of the deposition rate in Table 6, It can be seen that the deposition rate and target power of sample I5 and sample E4 are higher than those of sample E8 and sample C5. In Table 6, the deposition rate is reduced from I5 9.42 nm/min, E4 9.03 nm/min to E8 8.35 nm/min, and C5 8.08 nm/min. The increase in sputtering power will relatively increase the deposition rate of the film and thus promote grain refinement. In summary, the formation of nanocrystals is promoted by the combination of various factors such as slow diffusion effect, deposition rate, sputtering power, etc.

### 3.4. Surface Roughness

The AFM surface topography of samples E8, I5, C5, and E4 is shown in Figure 8. Figure 8 shows a distinct flake structure on the surface of the alloy film. This is consistent with the topography results presented in Figure 7. The scanning height is 0–130nm, and the scanning range is in the area of 5 × 5 μm. Due to the different atomic radii of the elements, there is a certain variation range of roughness Ra, about 11–16 nm, and the film is relatively smooth overall. The sample with the lowest Zr atom ratio, E8, had the lowest roughness of 11.88 nm, while sample E4 had the highest surface roughness of 15.94 nm due to the mixing of more Zr elements with large atomic radii. Therefore, we believe that the surface roughness of RMPAs is mainly affected by the atomic radius of the element, and as the content of elements with large atomic radii increases, the roughness would be higher. The following two factors also affect the surface roughness of the film. First, the mask plate used in this work caused a blocking effect on the sputtering particles, thereby affecting the deposition uniformity of the sample library. Second, the surface roughness is affected by the deposition rate of the sample [45]. The roughness of E8 is significantly lower than that of E4, and the deposition rate of E8 in Table 6 is 8.35 nm/min, which is significantly lower than that of E4 at 9.03 nm/min.

### 3.5. Mechanical Properties

The load–displacement curves of the ZrMoTaW film on the Si substrate are shown in Figure 9. In the two illustrations in Figure 9, it can be observed that there is a slight “pop-in” phenomenon near 125 nm and 180 nm, but the load changes evenly with the overall displacement. The load–displacement curves of the TaNbHfZr thin films obtained by nanometer indentation by Song et al. showed an obvious “pop-in” phenomenon at the displacements of 40 nm and 65 nm, respectively, and it was caused by uneven deformation triggered by partial crystallization [46].

Figure 10 summarizes the hardness and elastic modulus of the ZrMoTaW RMPEA gradient film. For D11 Zr_11_Mo_11_Ta_25_W_53,_ the hardness can reach 20 Gpa, and the elastic modulus exceeds 300 Gpa. Compared with the refractory metals reported in Table 7 [10,36,46,47,48,49,50,51,52], the ZrMoTaW RMPEA thin film can reach much higher hardness than other materials. We attribute the excellent mechanical properties mainly to: (1) The composition distribution of ZrMoTaW RMPEA film is locally homogeneous. The four-target co-sputtering forms a locally evenly distributed chemical composition, as shown in Figure 4, which reduces the defect of the film and improves the mechanical properties of the film. (2) ZrMoTaW RMPEA films inherit the excellent properties of high strength and hardness of nanocrystals [53,54]. The structure of the thin film is nanocrystalline, and the grain size does not exceed 15 nm. (3) The (110) preferential orientation is conducive to improving the hardness of ZrMoTaW RMPEA film [55,56]. It can be seen from the XRD pattern that the sample Zr_11_Mo_11_Ta_25_W_53_ has a much higher peak strength than other components, and its hardness is also much higher than that of other components. (4) The surface roughness affects the hardness of the film. As Jiang et al. discovered in their exploration of the effect of film surface roughness on the nanoindentation experiments, the nano hardness of rough films is generally lower than the predicted value of smooth films [57]. The peak strengths of E8 and E4 in this work are almost identical, but the roughness of E4 is significantly greater than that of E8. In Figure 10, E4 has the lowest hardness, and the hardness of E8 is significantly greater than that of E4. This is consistent with the reported experimental results of Jiang et al. The large surface roughness of the film means that the size of the micro-holes between the particles increases, which affects the density of the film and leads to a decrease in hardness [42,57,58,59]. (5) Solid solution strengthening significantly affects the hardness of multi-component alloy films. Lattice mismatch can seriously affect the mechanical properties of the alloy, in addition to being related to the phase stability described in Section 3.1 [2,60,61]. Due to the variety of elements, HEAs inevitably produce lattice distortion. A large number of dislocations are generated during deformation, and the interaction of these dislocations with the local stress field of the solute atoms causes the MPEAs to produce solid solution strengthening. The interaction forces can be inferred by Equation (9):(9)$$F={\mathrm{Gb}}^{2}\delta ={\mathrm{Gb}}^{2}\left[{\beta \delta }_{a}+\delta_{G}\right]$$
where G represents the shear modulus of the alloy, b is the magnitude of the Burgers Vector, β is a constant, the size mismatch $\delta_{a}$=$\frac{1}{a}\frac{da}{dc}$, the modulus mismatch δ_G_ = $\frac{1}{G}\frac{dG}{dc}$, and G is the shear modulus of the alloy. The constant β is related to the spiral dislocation and blade dislocation and local stress field caused by solute atoms [62,63]. It has been reported that β is 2–4 for spiral dislocations and β ≥ 16 for blade dislocations [62]. Since the alloy studied in this paper shows BCC structure, the spiral dislocation and the blade dislocation may exist at the same time, so the value of β is 9 [16].

The contribution of lattice distortion to solid solution strengthening can be assessed by quantitative calculations. In multi-principal alloys of single-phase solid solutions, lattice distortion and modulus distortion are the main causes of solid solution strengthening, which can be expressed in $\Delta \sigma_{s}:$(10)$$\Delta \sigma_{s}=\Delta \sigma_{a}+\Delta \sigma_{G}$$
where $\Delta \sigma_{a}$ and $\Delta \sigma_{G}$ are the contributions of lattice distortion and modulus distortion to solid solution strengthening.
(11)$$\Delta \sigma_{a}=AG\delta_{ai}{}^{4/3}c_{ai}{}^{2/3}$$
(12)$$\Delta \sigma_{G}=AG\delta_{Gi}{}^{4/3}c_{Gi}{}^{2/3}$$
where *A* is a dimensionless constant, *G* is the shear modulus of the alloy, *c* is the solute concentration,$\;\delta_{ai}$ and $\delta_{Gi}$ are average atomic size mismatch and average atomic modulus mismatch, respectively [64].$\;\delta_{ai}$ and $\delta_{Gi}$ can be defined via Equations (13) and (14) [63,64].
(13)$$\delta_{ai}=\frac{9}{8}\sum c_{j}\delta_{a_{ij}}$$
(14)$$\delta_{Gi}=\frac{9}{8}\sum c_{j}\delta_{G_{ij}}$$
where *c_j_* is the atomic fraction of *j*th element in the alloy, *r_i,_* and *r_j_* are the atomic radius of *i*th and *j*th elements, and G_i_ and G_j_ are the shear modulus of ith and jth elements, respectively, and $\delta_{a_{ij}}=$ 2(*r_i_* − *r_j_*)/(*r_i_* + *r_j_*), $\delta_{G_{ij}}=$ 2(*G_i_* − *G_j_*)/(*G_i_* + *G_j_*). To facilitate the assessment of the contribution of distortion to HEAs as well as later studies, the value of *c* is specified as 0.25.

For ease of calculation, the component ratio is selected as an equivalent component Zr_25_Mo_25_Ta_25_W_25_. The calculated atom size distortion and atomic modulus distortion are shown in Table 8. Table 9 shows lattice distortion and modulus distortion near each element in Zr_25_Mo_25_Ta_25_W_25_ solid solution alloy. The data in Table 8 show that the combined atomic size difference between Mo and W is small, as $\delta_{a_{ij}}\approx$ 0.01. Compared to the combination with the Zr element, the difference in atomic size between W and Mo is smaller. The size difference between Zr and other elements is as small as $\delta_{a_{ij}}\approx$ 0.08 and up to $\delta_{a_{ij}}\approx$ 0.13. The lattice distortion near the elements shown in Table 9 is consistent with the above data. The lattice distortion near the Zr element is the greatest,$\;\delta_{a_{ij}}\approx$ 0.096. As is expected, the elements with a small radius such as Mo and W produce almost the same local tensile strain,$\;\left|\delta_{a_{i}}\right|$~0.05, while the element with large radius Zr and Ta, produce local compression,$\;\delta_{a_{i}}$~0.08–0.096. This local strain is consistent with the lattice distortion near elements in TaNbHfZrTi alloys studied by Senkov et al. [16].

In Table 8, the values of modulus difference between elements range widely, from $\delta_{Gi}$ = 0.07 for the Mo–W atom pair to 1.79 for the Zr–W atom pair. The strongest shear modulus effect of Zr atoms with other elements is that $\delta_{G_{ij}}$ values range from 0.71 for Zr-Ta to 1.17 for Zr-W, while Mo-W, Mo-Ta, and Ta-W pairs resulted in smaller $\delta_{G_{ij}}$ values of 0.07, 0.52, and 0.59, respectively. The modulus distortion near the specific element calculated in Table 9 shows that the highest value $\delta_{G_{Zr}}$ = 0.844 is displayed near the Zr atom. Essentially, the Zr interaction produces greater deviations in local forces than the other three elements, resulting in a large effective modulus mismatch. Senkov et al.’s study of Ta has a similar contribution to Zr in this work and they hypothesized that the contribution is mainly derived from Ta. This work assumes that the mismatched contributions are mainly due to Zr. The dimensionless constants A in Equations (11) and (12) take 0.04 [64], and experimentally obtained Er of ZrMoTaW is ~150 GPa, thus the shear modulus G is 56.82 GPa (calculated by G = E_r_/[2(1 + v)]), taking v as 0.32 here). The equiatomic of ZrMoTaW c_i_ is 0.25. The resulting atomic size difference contribution is 38.55 MPa and the contribution of the shear modulus is 719.71 MPa. Thus, the dominant contribution to the strengthening is the modulus distortion. The reinforcement effect of this experiment is the same as that of the reported TaNbHfZrTi alloy system [16].

Hardness and modulus have an important influence on wear behavior [65,66]. As mentioned above, the ZrMoTaW film prepared in this work has extremely high hardness, which indicates that the film may have excellent wear resistance. Studies have shown that the ratio of hardness to elastic modulus (H/Er) indicates the ability of the material to resist elastic strain before failure [67], and H^3^/Er^2^ is a plastic deformation factor, which indicates the ability to resist plastic flow [68]. The H/Er and H^3^/Er^2^ values are shown in Figure 11, and the higher the ratio of these two values, the better the wear resistance [69]. It can be seen that the ratio of D11 is significantly higher than that of other sample libraries, which means that the wear resistance of D11 is better than that of other areas. This means that ingredients with greater stiffness, modulus, and abrasion resistance can be explored near the area. The highest values of H/Er and H^3^/Er^2^ of ZrMoTaW RMPA reached 0.066 and 0.086, respectively, which are significantly higher than ordinary alloys such as ZrCrAlN [70], AlCrSiN [71]. For traditional aluminum-based alloy films, the maximum value of H^3^/Er^2^ is still lower than 0.02GPa. As a result, the ZrMoTaW RMPEA exhibits excellent protection potential compared to many conventional films. Moreover, the high throughput proposed in this study can quickly identify the property-changing trend brought about by the influence of different elements. It exhibits better hardness and abrasion resistance on the side with high W content.

In this work, a thin film library of ZrMoTaW systems with multi-directional gradient components was prepared by multi-gradient deposition combination and porous masking. The results show that:

(1)By adjusting the power and controlling the distance between the sample and the target, changes in the spatial gradient and composition range can be controlled. The composition of each element changed uniformly in a gradient from different directions. Local uniformity in the element distribution can be observed with the help of SEM surface scanning. This facilitates screening ingredients by demand and exploring the effects of elemental variations.(2)The XRD atlas shows that the ZrMoTaW alloy film is a single-phase solid solution with a BCC structure. According to the Hume-Rothery rule, the theoretical parameters of ZrMoTaW alloy thin films Ω ≥ 1.1, δ ≤ 6.6%, VEC ≤ 6.87, which meet the conditions for the formation of the BCC solid-solution phase structure. And ZrMoTaW RMPEAs have a high mixing entropy.(3)The ZrMoTaW thin films have a nanocrystalline structure with grain sizes of 10-20 nm. The surface morphology of the film shows a flake structure, and the cross-section shows a cylindrical structure. The diffusion of phase separation in HEA is slow. Higher sputtering and deposition rates also refine grains. Multiple factors work synergistically to promote grain refinement.(4)The ZrMoTaW thin films have excellent mechanical properties. The component with the highest hardness performance in this film is Zr_11_Mo_11_Ta_25_W_53_. The maximum hardness measured is up to 20 GPa, and the modulus of elasticity exceeds 300 GPa. Through quantitative calculations, the main contribution of the film reinforcement produced in this experiment comes from modulus distortion, and it has excellent wear resistance. The H/Er and H^3^/Er^2^ of ZrMoTaW RMPEA reached 0.066 and 0.086, respectively.

The structure of ZrMoTaW films will be studied further. The study will be based on the properties of bulk alloys with high W content. Ingredients that are expected to have excellent properties in the film can also be found in the block.

## Figures and Tables

**Figure 1 materials-15-08546-f001:**
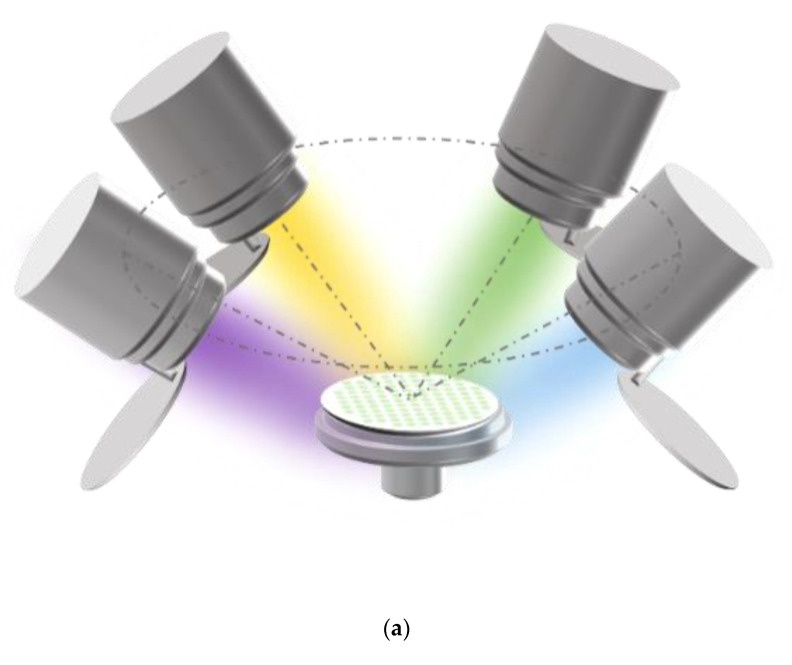
(**a**) Schematic diagram of the four-target co-sputtering. (**b**) The dimensional design of the aluminum mask plate.

**Figure 2 materials-15-08546-f002:**
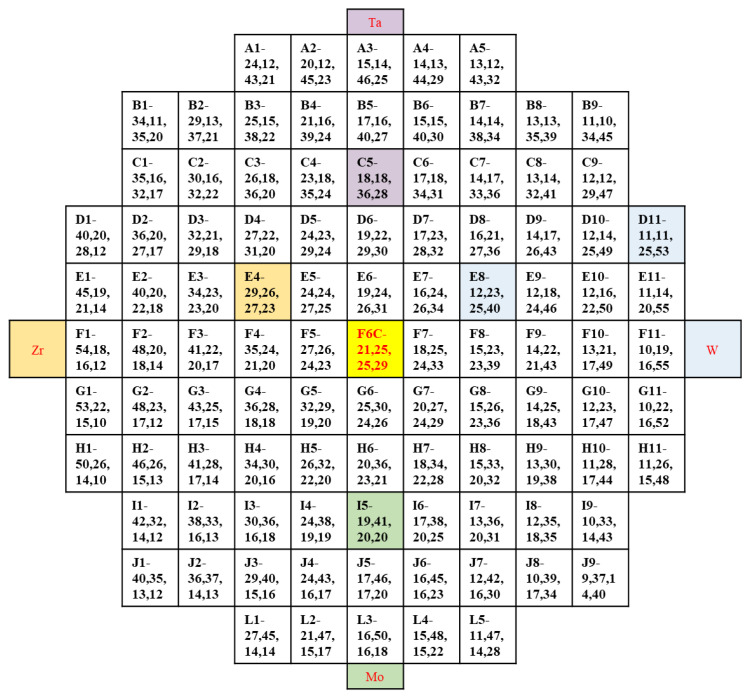
The composition of the 101 sample cells in the film.

**Figure 3 materials-15-08546-f003:**
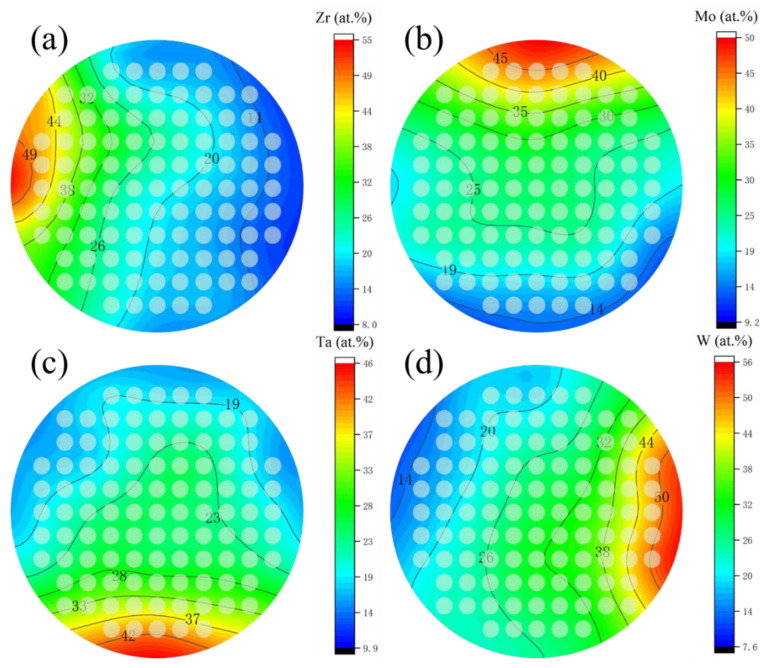
Composition gradients of (**a**) Zr, (**b**) Mo, (**c**) Ta, and (**d**) W of thin film.

**Figure 4 materials-15-08546-f004:**
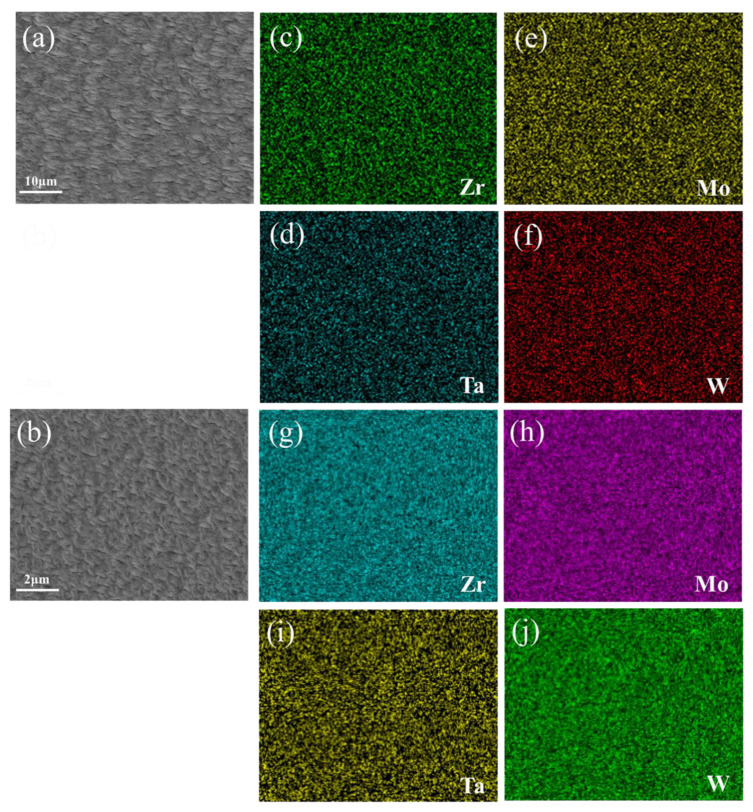
(**a**,**b**) SEM images of surface morphology of F6C and D11 thin film and (**c**–**j**) EDS elemental maps of region shown in (**a**,**b**).

**Figure 5 materials-15-08546-f005:**
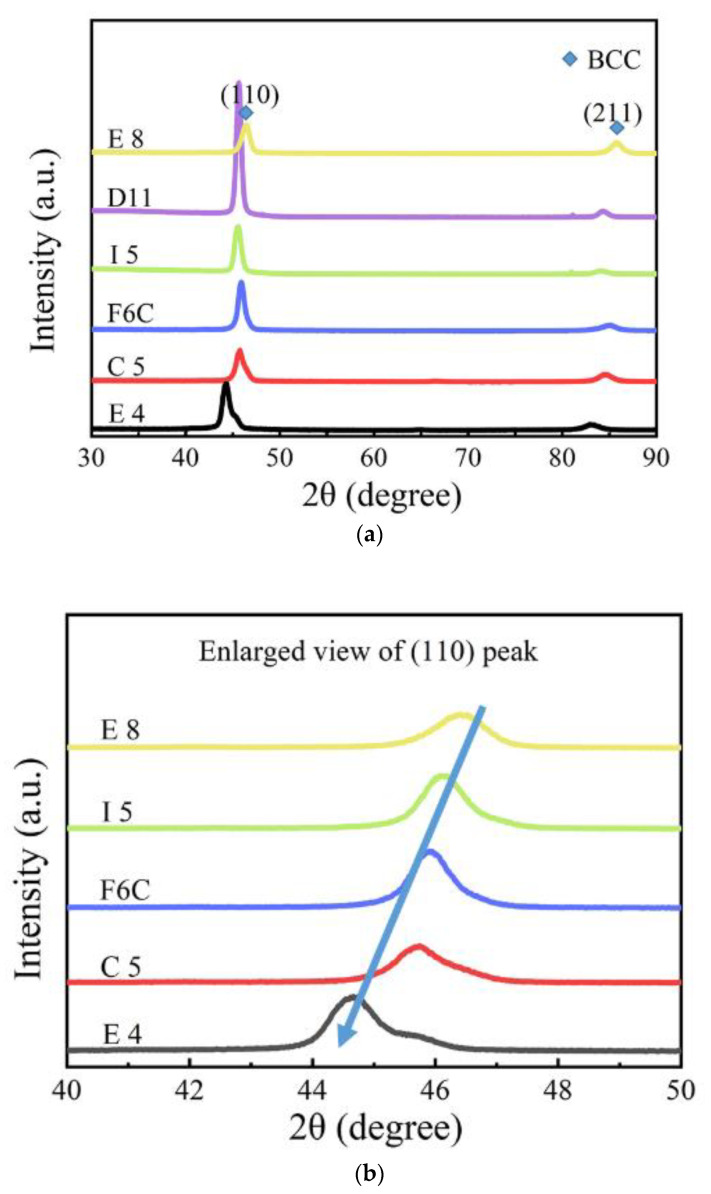
(**a**) XRD patterns of Zr-Mo-Ta-W RMPEA thin films, and (**b**) enlarged view of samples E8, I5, F6C, C5, and E4 between 40° and 50°.

**Figure 6 materials-15-08546-f006:**
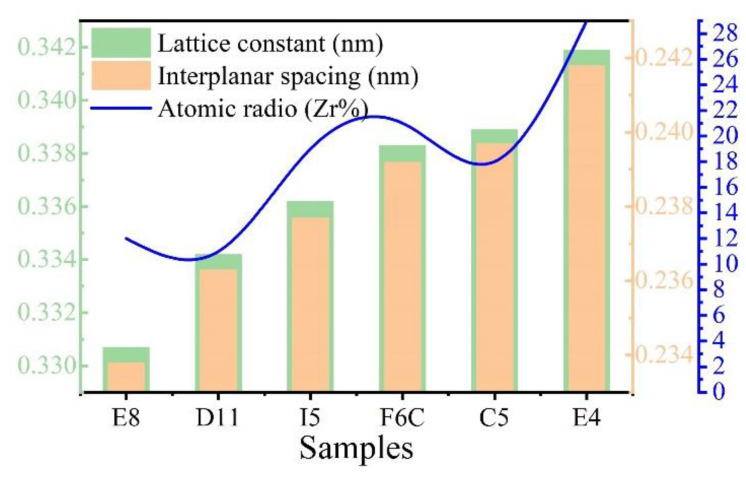
The lattice constant, interplanar spacing, and atomic radio (Zr%) of Zr-Mo-Ta-W RMPEA thin films.

**Figure 7 materials-15-08546-f007:**
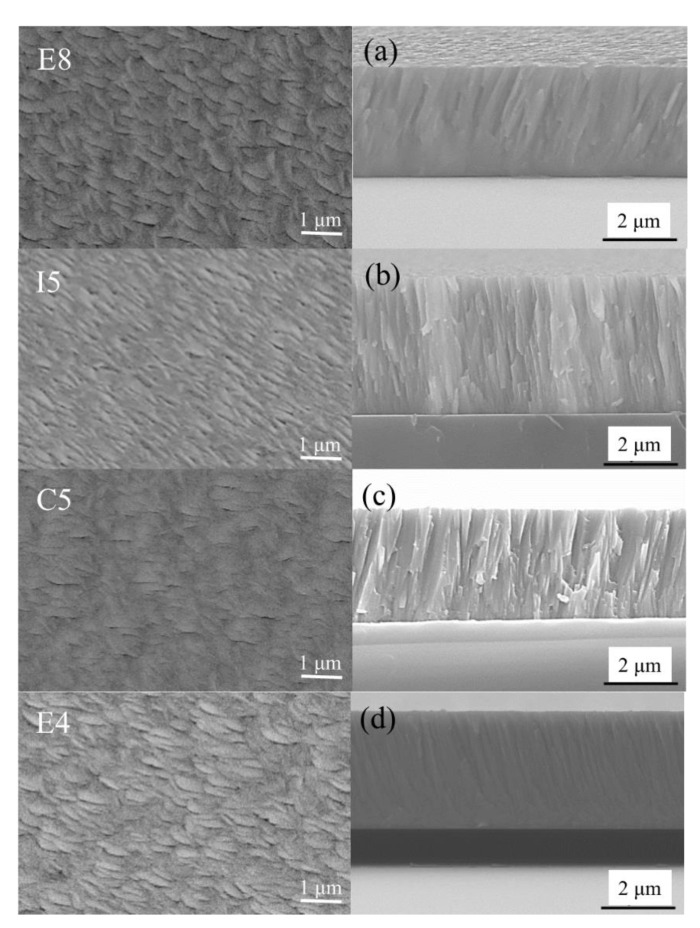
SEM plan-view morphology and cross-sectional of ZrMoTaW RMPEA thin films: E8, (**a**) Zr_12_Mo_23_Ta_25_W_40_; I5, (**b**) Zr_19_Mo_41_Ta_20_W_20_; C5, (**c**) Zr_18_Mo_18_Ta_36_W_28_; E4, (**d**) Zr_29_Mo_27_Ta_26_W_18_.

**Figure 8 materials-15-08546-f008:**
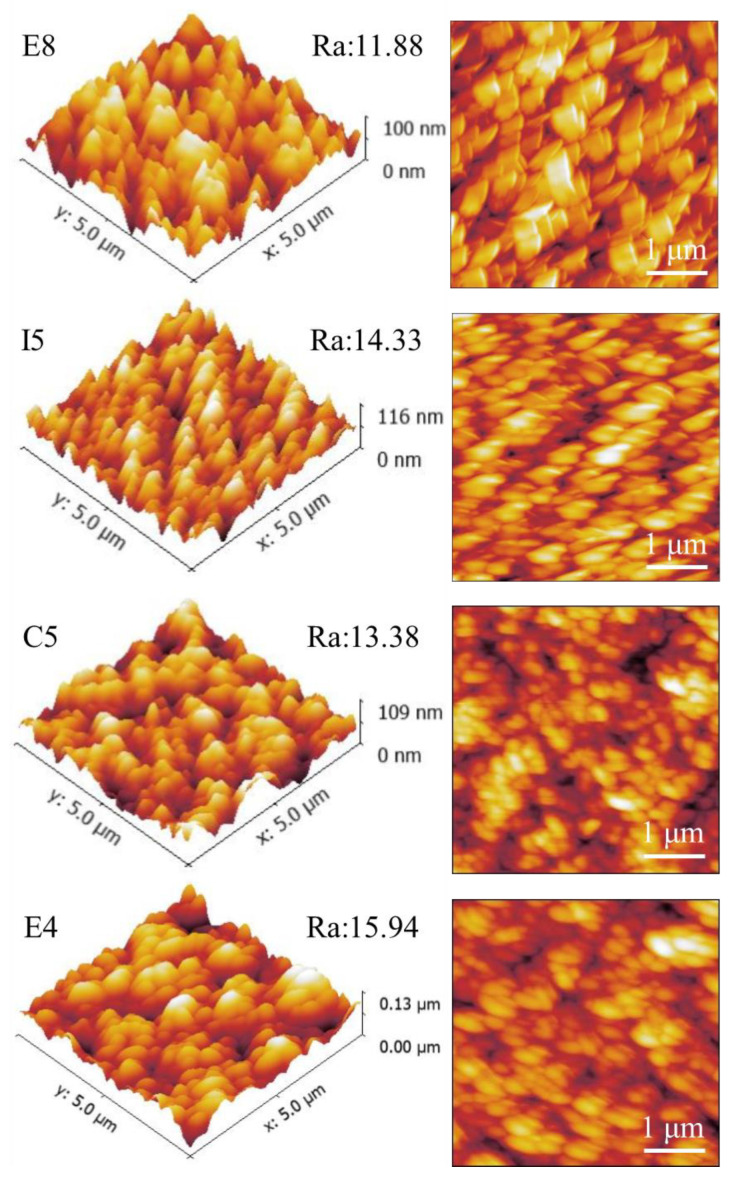
AFM surface roughness of ZrMoTaW RMPEA thin films: E8 Zr_12_Mo_23_Ta_25_W_40_; I5 Zr_19_Mo_41_Ta_20_W_20_; C5 Zr_18_Mo_18_Ta_36_W_28_; E4 Zr_29_Mo_27_Ta_26_W_18_.

**Figure 9 materials-15-08546-f009:**
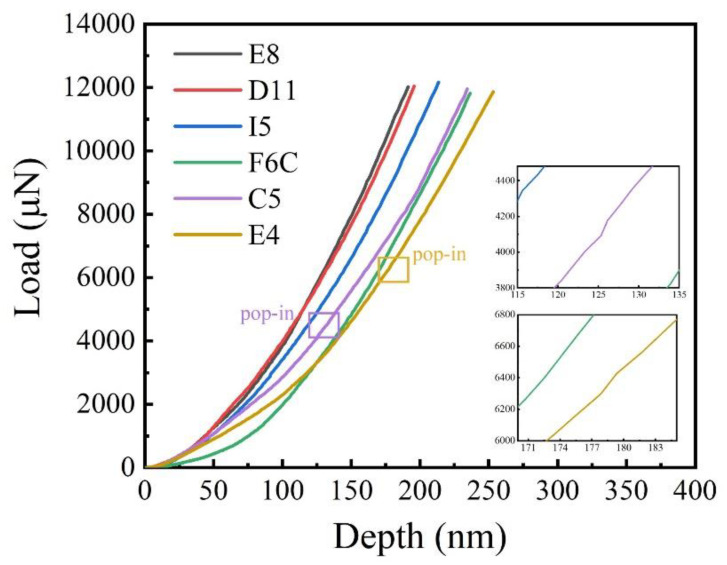
Typical load–displacement curves of ZrMoTaW film and the enlarged inset shows the “pop-in” phenomenon for Zr_18_Mo_18_Ta_36_W_28_ and Zr_29_Mo_27_Ta_26_W_18_.

**Figure 10 materials-15-08546-f010:**
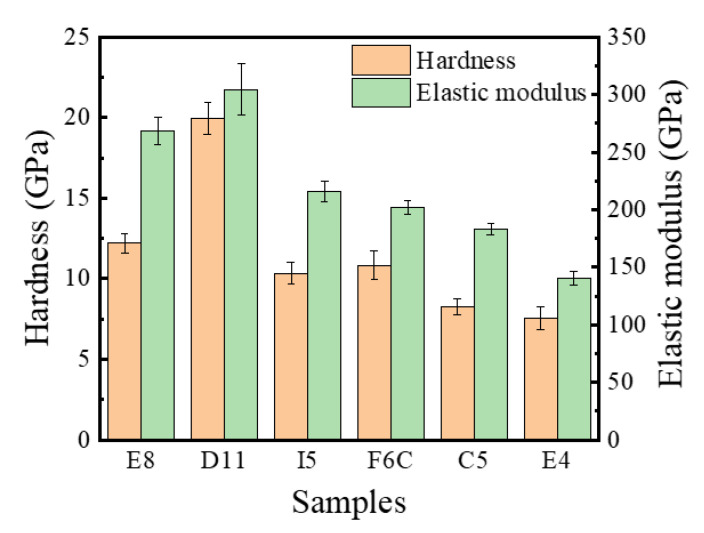
Nano hardness and elastic modulus ZrMoTaW RMPEA thin films: E8, D11, I5, F6C, C5, E4.

**Figure 11 materials-15-08546-f011:**
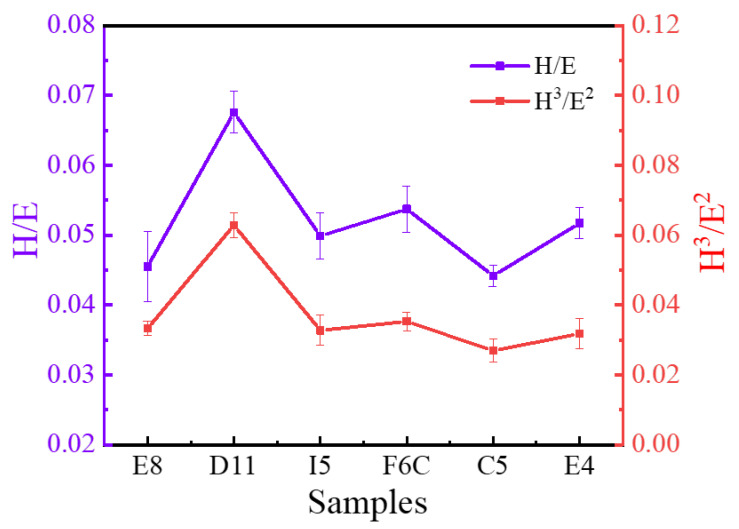
Elastic strain to failure, H/Er, and resistance to plastic deformation, H^3^/E^2^, of ZrMoTaW RMPEA thin films.

**Table 1 materials-15-08546-t001:** Experimental parameters of the ZrMoTaW RMPA films.

Samples	Composition (at. %)				Working Gas	Working Pressure (Pa)
	Zr	Mo	Ta	W		
E8	12	23	25	40	Ar	0.4
D11	11	11	25	53		
I5	19	41	20	20		
F6C	21	25	25	29		
C5	18	18	36	28		
E4	29	27	26	18		
Target power (W)	35	30	19	20		

**Table 2 materials-15-08546-t002:** Structure, atomic radius, and lattice constant of Zr, Mo, Ta, and W elements in the ZrMoTaW system [10,34,35,36].

	Element			
	Zr	Mo	Ta	W
Structure	bcc	bcc	bcc	bcc
Radius (nm)	0.160	0.140	0.148	0.141
Lattice constant (nm)	0.3695	0.3233	0.3349	0.3256

**Table 3 materials-15-08546-t003:** Thermodynamic parameters of the ZrMoTaW RMPEA thin films with different compositions.

Samples	Composition	Atomic Size Difference (%)	Mixing Enthalpy (kJ·mol^−1^)	Mixing Entropy (kJ·mol^−1^)	Theoretical Melting Point (K)	Thermodynamic Parameter (Ω)	Valence Electron Effect (VEC)
		δ (%)	ΔH (kJ·mol^−1^)	ΔS (kJ·mol^−1^)	Tm (K)	Ω	VEC
E8	Zr_12_Mo_23_Ta_25_W_40_	4.11	−5.98	10.85	2657	4.82	5.51
D11	Zr_11_Mo_11_Ta_25_W_53_	5.49	−6.32	9.72	2764	4.25	5.53
I5	Zr_19_Mo_41_Ta_20_W_20_	5.21	−5.54	11.01	2429	4.83	5.42
F6C	Zr_21_Mo_25_Ta_25_W_29_	4.32	−6.10	11.47	2503	4.71	5.33
C5	Zr_18_Mo_18_Ta_36_W_28_	3.90	−5.93	11.15	2564	4.82	5.28
E4	Zr_29_Mo_27_Ta_26_W_18_	4.45	−5.57	11.40	2531	5.18	5.83

**Table 4 materials-15-08546-t004:** Texture coefficient of (110) and (211) planes.

	E8	D11	I5	F6C	C5	E4
(110)	0.91	1.77	1.56	1.52	1.15	1.48
(211)	1.08	0.23	0.44	0.48	0.85	0.51

**Table 5 materials-15-08546-t005:** Average atomic radium, lattice constant, interplanar spacing, and diffraction angle (110).

Samples	Composition	Average Atomic Radium (nm)	Lattice Constant (nm)	Interplanar Spacing (nm)	Diffraction Angle (θ)
		$\bar{r}$ (nm)	a_n_ (nm)	d*_hkl_* (nm)	(θ)
E8	Zr_12_Mo_23_Ta_25_W_40_	0.1432	0.3307	0.2338	22.51
D11	Zr_11_Mo_11_Ta_25_W_53_	0.1447	0.3342	0.2363	22.26
I5	Zr_19_Mo_41_Ta_20_W_20_	0.1456	0.3362	0.2377	22.12
F6C	Zr_21_Mo_25_Ta_25_W_29_	0.1465	0.3383	0.2392	21.97
C5	Zr_18_Mo_18_Ta_36_W_28_	0.1468	0.3389	0.2397	21.73
E4	Zr_29_Mo_27_Ta_26_W_18_	0.1481	0.3419	0.2418	21.73

**Table 6 materials-15-08546-t006:** Mean grain size, thickness, and deposition rate of ZrMoTaW RMPEA thin films.

Samples	Composition	Grain Size (nm)	Thickness (μm)	Deposition Rate (nm/min)
E8	Zr_12_Mo_23_Ta_25_W_40_	11.94	3.005	8.35
I5	Zr_19_Mo_41_Ta_20_W_20_	11.01	3.392	9.42
C5	Zr_18_Mo_18_Ta_36_W_28_	10.90	2.909	8.08
E4	Zr_29_Mo_27_Ta_26_W_18_	10.82	3.252	9.03

**Table 7 materials-15-08546-t007:** The hardness values of refractory metals, RMPEAs blocks, and RMPEAs thin films.

Metals	Hardness (GPa)		Reference
Bulk	Zr	9	[47]
	Mo	6	[48,49]
	Ta	2	[48,50]
	W	8	[49,50]
	TaNbHfZr	3.58	[36]
	NbMoTaW	4.46	[10]
	NbMoTaWV	5.25	[10]
Films	NbMoTaW	13.5	[63]
	TaNbHfZr	15.3	[46]
	TiVCrAlZr	8.2	[52]
	This work	20	

**Table 8 materials-15-08546-t008:** Atomic size difference $\delta_{a_{ij}}$ and modulus difference $\delta_{G_{ij}}$ with bold numbers, of the alloying.

$$\delta_{G_{ij}}\;/\;\delta_{a_{ij}}$$	Zr	Mo	Ta	W
Zr	0	−0.13	−0.08	−0.13
Mo	**1.13**	0	0.06	0.01
Ta	**0.71**	**−0.52**	0	−0.05
W	**1.17**	**0.07**	**0.59**	0

**Table 9 materials-15-08546-t009:** Calculated lattice distortion $\delta_{G_{i}}$ and modulus distortion $\delta_{G_{i}}$ near each element in the BCC lattice of the ZrMoTaW alloy.

$\delta_{G_{i}}\;/\;\delta_{a_{i}}$	Zr	Mo	Ta	W
$\delta_{a_{i}}$	0.096	−0.056	0.008	−0.047
$\delta_{G_{i}}$	−0.844	0.446	0.113	0.512

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
