# Peer review of "High-Throughput Preparation and Characterization of ZrMoTaW Refractory Multi-Principal Element Alloy Film"

_materials, 2022, doi:10.3390/ma15238546_

Round 1

Reviewer 1 Report

The paper titled High-throughput preparation and characterization of 2 ZrMoTaW refractory multi-principal element alloy film addressed very interesting topic and has enough data to confirm the novelty. However, few minor corrections are required before the final

1.     Figure 1 has a Chinese word need to be translated to English

2.     Add more quantitative and qualitative results in the abstract

3.     Highlight the main findings in the conclusion

Reviewer 2 Report

1- Similarity is 36% to published papers. Please reduce the similarity percentage to 25% or lower.

2- Include more main results in abstracts.

3- In the introduction explain why Zr-Mo-Ta-W is chosen for study.

4- Check capitallization, space between letters and spelling for the whole manuscript. 

5- Check citations, authors name.

6- Many equations were mentioned, but, it is not clear how it relates to the results and discussion presented. 

7- The overall presentation of the results and discussion can be improved. As an example, explain the characterization and the properties collected. Why it is important to quantify these properties and how does it reflect on the material properties. Relate the results from one part to another if a relationship can be established. That way the discussion can be more meaningful.

8- The results also contain several results from equation calculations. Explain how each and every parameter were obtained and find a way to prove that your calculations are correct. Maybe comparison to other established data from other literature.

9- Figure 4 is confusing. From which sample it is taken? Why it is required? Is it necessary?

10- Table 2, measured or calculated yourself? or taken from references? if calculated, please explain how. if taken from other sources, please cite.

11- Check references writing format. 

Reviewer 3 Report

Wang et al. have presented the manuscript titled: High-throughput preparation and characterization of ZrMoTaW refractory multi-principal element alloy film. Overall presentation of the work is good, but there are few suggestions which I think are necessary to explain before publication.

1.      Abstract of the manuscript is so limited. Authors have strongly focused on the fabrication technique in abstract. I suggest the authors to revise the abstract and add your achieved results (values) in it.

2.      I suggest the authors to describe the W content in the abstract, what variation in the high content they used must be elaborated in abstract to make the manuscript appealing for readers.

3.      Please revise the whole manuscript carefully, there are so many mistakes of spacing, spellings and grammatical, please correct them.

4.      Problem statement in the introduction section is weak. I suggest the authors to elaborate why they have performed this research work and what was lacking in the previous studies which have compelled the authors to carry out this research work.

5.      In the experimental sections authors have used 4 targets, were these targets prepared by authors or were they purchased? Please describe the specifications of Manufacturer Company and country.

6.      In Figure 4, is this elemental mapping performed for 10 µm sample or for 2µm sample? And what concentration sample is this?

7.      N.A. Khan et al. demonstrated EDS mapping of films, I suggest the authors to check the EDS for their best sample and add it in Figure 4.

8.      Authors have performed the XRD study of fabricated material. But here the information about structural analysis is so less. What is the structure of the material? What is its space group? If authors have checked the lattice parameters, from which structure they have used as reference?

9.      I suggest the authors to describe their claim of structure by PDF Card# previously reported.

Reviewer 4 Report

The article of the team of authors is devoted to the study of the properties of films made of four-component refractory alloys with several basic elements (RMPEA) with a high content of W. In general, the presented work has a certain level of novelty and practical significance, and the obtained dependences can later be used in practice in similar developments. The article has a good structure, as well as a large amount of illustrative material, which quite well reflects the presented results and the obtained dependencies. In general, the article corresponds to the subject of the journal, and can also be accepted for publication after the authors answer all the questions.

1. In the abstract, the authors should provide more details about the research being conducted and its novelty.

2. The authors should give more explanations for such a strong difference in the surface morphology and the degree of roughness of the samples obtained under different conditions. Roughness measurement profiles should also be given.

3. The results of strength properties should be supplemented with these changes in surface hardness in depth.

4. According to the elemental composition data, the homogeneity of the coating of all four elements is the same and uniform, the authors should give more explanations of the results.

5. The authors should give an explanation about such a strong distortion of the crystal lattice for various samples, and also what is the reason for such a change in the structural parameters.

6. In conclusion, the authors should reflect further plans for research.

7. For all observed experimentally obtained parameters, the values ​​of standard deviations should be given.

Round 2

Reviewer 2 Report

Changes has been made according to the previous comments. 

Reviewer 4 Report

The authors answered all questions, the article can be accepted for publication.
